# Application of Data Sensor Fusion Using Extended Kalman Filter Algorithm for Identification and Tracking of Moving Targets from LiDAR–Radar Data

Oscar Javier Montañez [1], Marco Javier Suarez [2] and Eduardo Avendano Fernandez [1,*]

[1] School of Electronic Engineering, Pedagogical and Technological University of Colombia, Sogamoso 152210, Colombia; oscarjavier.montanez@uptc.edu.co

[2] School of Systems and Computing Engineering, Pedagogical and Technological University of Colombia, Sogamoso 152210, Colombia

[*] Correspondence: eduardo.avendano@uptc.edu.co

**Abstract:** In surveillance and monitoring systems, the use of mobile vehicles or unmanned aerial vehicles (UAVs), like the drone type, provides advantages in terms of access to the environment with enhanced range, maneuverability, and safety due to the ability to move omnidirectionally to explore, identify, and perform some security tasks. These activities must be performed autonomously by capturing data from the environment; usually, the data present errors and uncertainties that impact the recognition and resolution in the detection and identification of objects. The resolution in the acquisition of data can be improved by integrating data sensor fusion systems to measure the same physical phenomenon from two or more sensors by retrieving information simultaneously. This paper uses the constant turn and rate velocity (CTRV) kinematic model of a drone but includes the angular velocity not considered in previous works as a complementary alternative in Lidar and Radar data sensor fusion retrieved using UAVs and applying the extended Kalman filter (EKF) for the detection of moving targets. The performance of the EKF is evaluated by using a dataset that jointly includes position data captured from a LiDAR and a Radar sensor for an object in movement following a trajectory with sudden changes. Additive white Gaussian noise is then introduced into the data to degrade the data. Then, the root mean square error (RMSE) versus the increase in noise power is evaluated, and the results show an improvement of 0.4 for object detection over other conventional kinematic models that do not consider significant trajectory changes.

**Keywords:** data sensor fusion; extended Kalman filter; lidar; radar

## 1. Introduction

In surveillance and monitoring systems, the use of unmanned aerial vehicles (UAVs), such as drones or mobile vehicles, provides advantages in terms of access to the environment for exploration like augmented range, maneuverability, and safety due to their omnidirectional displacement capacity. These tasks must be performed autonomously by capturing information from sensors in the environment at scheduled or random points at specific times and areas. The collected data present errors and uncertainties that make object recognition difficult and depend on the resolution of the sensors for detection and identification. Data acquisition resolution can be improved by integrating sensor data fusion systems to measure the same physical phenomenon by capturing information from two or more sensors simultaneously and applying filtering or pattern recognition techniques to obtain better results than those obtained with only one sensor

Sensor data fusion consists of different techniques, inspired by the human cognitive ability to extract information from the environment by integrating different stimuli. In the case of sensor fusion, measurement variables are integrated through a set of sensors,

often different from each other, that make inferences that cannot be possible from a single sensor [1].

The fusion of Radar (Radio Detecting and Ranging) and LiDAR (Light Detection and Ranging or Laser Imaging Detection and Ranging) sensor data presents a better response considering two key aspects: (i) the use of two coherent systems that allow an accurate phase capture and (ii) the improvement in the extraction of data from the environment, with the combination of two or more sensors arranged on the mobile vehicle or UAV [1,2]. This integration allows the error to be decreased in the detection of objects in a juxtaposition relationship by determining the distances through the reflection of radio frequency signals in the Radar case and through the reflection of a light beam (photons) for the case of the LiDAR sensor, generating a double observer facing the same event, in this case, the measurement of proximity and/or angular velocity [3,4]. Thus, the choice of Radar and LiDAR sensors requires special care, mainly about technical characteristics and compatibility [5,6], coherence in range, and data acquisition. The above allows a complementary performance to be achieved with its associated element in data fusion, facilitating a better understanding of the three-dimensional environment that feeds the data processing system integrated into the UAV [7] or at a remote site.

A proper sensor fusion of LiDAR and Radar data must rely on the use of estimators to achieve higher consistency in the measurements to mitigate the uncertainties by using three parameters: Radar measurements, LiDAR measurements, and Kalman filtering. This improves the estimation of the measured variable. The Kalman filtering technique allows the description of the real world using linear differential equations to be expressed as a function of state variables. In most real-world problems, the measurements may not be a linear function of the states of the system. However, applying extended Kalman filtering (EKF) techniques counteracts this situation by modeling the phenomenon using a set of nonlinear differential equations, $X_k$, which describe the dynamics of the system. The EKF allows "projecting" in time the behavior of the system to be filtered, with variables that are non-measurable but are calculable from the measurable variables. Then, by predicting the future data and their deviation concerning the measured data, the Kalman gain, $K_k$, is calculated, and it continuously adapts to the dynamics of the system. Finally, updating the matrix state $\bar{x}_k$ and the covariance matrix $P_k$ associated with the filtered system. This process is graphically described in Figure 1.

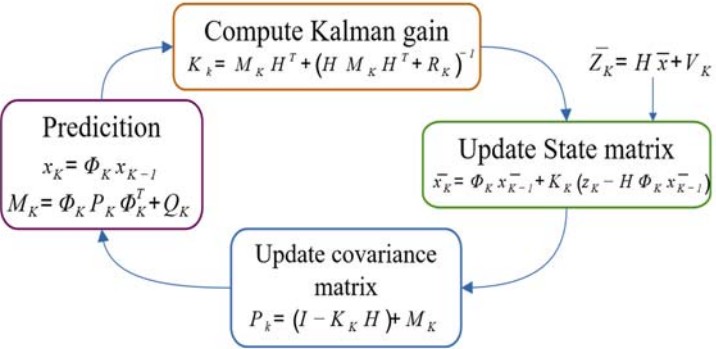

**Figure 1.** The schematic diagram for an extended Kalman filter.

In this work, sensor data fusion was performed for target tracking from a UAV, using an EKF and taking into consideration the results from data fusions performed in autonomous driving. The kinematic modeling Constant Turn Rate and Velocity (CTRV) [8] was taken as a reference, and this model includes in its description the angular velocity variable, provided by the Radar, a parameter that introduces an improvement in omnidirectional motion detection.

This paper shows the performance of an implementation of data sensor fusion using LiDAR and Radar through an EKF for the tracking of moving targets, taking into account

changes in their direction and trajectory, to generate a three-dimensional reconstruction when the information is captured from a UAV.

## 2. Dynamic Model of UAV

The UAV dynamics were obtained from the 2D CTRV model [8] for vehicle and pedestrian detection on highways. It is assumed that the possible movements of the elements around the UAV are not completely arbitrary and not holonomous, in which case there will be displacements in a bi-dimensional plane. The curvilinear model (CTRV) includes angular velocities and angular movements in its modeling, which allows a better description of the changes in the direction and velocity of an object in a linear model. The CTRV model is shown in Figure 2.

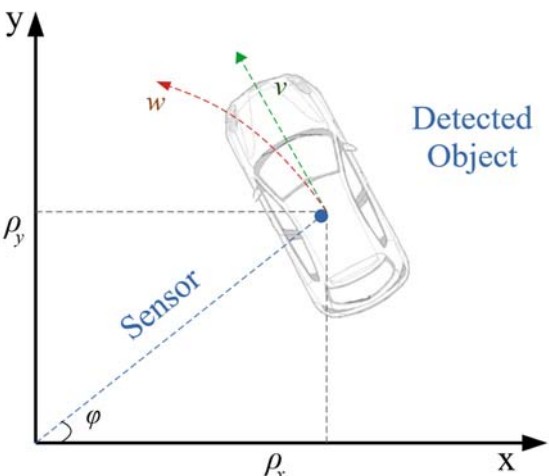

**Figure 2.** CTRV model for a moving object.

The velocity variable provides the system model the ability to calculate the target's lateral position variations for a correct prediction of the future position of the target, thus starting from initial positions $x$ and $y$ and projecting this location over time, defined as $x + \Delta x$ and $y + \Delta y$ for the target as shown in Figure 3.

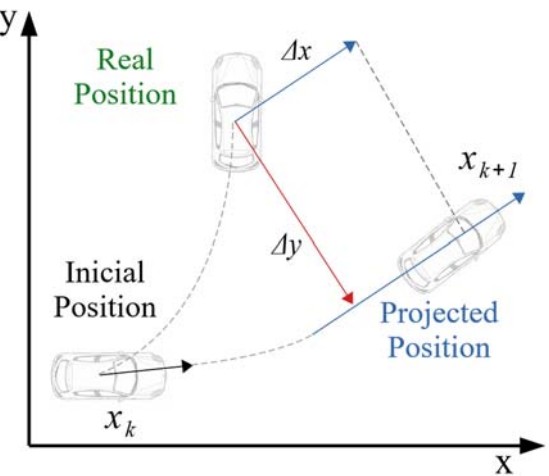

**Figure 3.** Position prediction through the CTRV model.

The CTRV model for the UAV system's moving target in the three-dimensional case determines the projection of the position of the target $x_{i+1}$ on the axis, starting from the values of the angular frequency $w$ and the angle $\theta$ [9–11] for $x_i$, and equally for $y_i$ and its position projection. Therefore, the variables of interest in the system are the position $x$

and $y$; these are calculated by modeling their projection through the frontal velocity $v$, the angle $\theta$ formed between the Radar and the target, the angular frequency $w$ of the target, and finally the angular frequency $w_d$ of the UAV. The set of state variables involved in the system is the following:

$$\overline{x} = [x, y, v, \theta, w, w_d] \tag{1}$$

The kinematic equations describing the change from an initial position of the UAV to a future position are as follows:

$$\overline{x}_{i+1} = x_i + \left[ v_{object} - v_{drone} \right] \cdot \Delta T \tag{2}$$

$$\overline{y}_{i+1} = y_i + \left[ v_{object} - v_{drone} \right] \cdot \Delta T \tag{3}$$

The state variables are the frontal velocity, the theta angle, the target angular velocity, and the angular velocity of the UAV.

$$\begin{aligned} v &= w \cdot \Delta T \\ \theta &= 0 \\ w &= 0 \\ w_d &= 0 \end{aligned} \tag{4}$$

Because the data sensor fusion operation is bidimensional, the CTRV model does not include motion in the position around the $z$-axis in its state variables. To maintain a bi-dimensional analysis, the UAV velocity [10] is projected as

$$v_x = V \cos \phi \tag{5}$$

$$v_y = V \sin \phi \tag{6}$$

In this way, $\phi$ represents the elevation of the UAV concerning the sensed target, this angle allows the velocities of the drone to be projected in the $xz$ plane, and the $x + \Delta x$ or $y + \Delta y$ to be determined, as shown in Figure 3, concerning the position prediction. To simplify the model and to have a congruence of the LiDAR and Radar models in the sensor data fusion, a data acquisition method is proposed in which the UAV only uses pitch (rotation on the lateral Y axis) and yaw (rotation on the vertical Z axis) movements, and their projection in a three-dimensional coordinate system. These motions are included in the CTRV model through the projection of the UAV velocity $v_d$, through the angles $\phi$ and $\theta$, as shown below.

$$v_d = \begin{bmatrix} V_{dx} \\ V_{dy} \\ V_{dz} \\ W_{dz} \end{bmatrix} = \begin{bmatrix} V_0 \cos \phi \cos \theta \\ V_0 \cos \phi \sin \theta \\ V_0 \sin \phi \\ 0 \end{bmatrix} \tag{7}$$

The difference between the estimated position and the actual position of the target is determined by the displacement generated by w and $\theta$, i.e., ($\Delta T \cdot w + \theta$) [8], so the space and velocity projections are also a function of these variations. The velocity equations are obtained from $x_i$ and $y_i$, which correspond to the first derivative, such that $v_{dx}$ and $v_{dy}$ are expressed as

$$\begin{aligned} \dot{x}_{i+1} &= \tfrac{v}{w} [sin(\Delta T \cdot w + \theta) - sin\theta] - v_{dx} \cos \theta \cos \phi \\ \dot{y}_{i+1} &= \tfrac{v}{w} [\cos \theta - \cos(\Delta T \cdot w + \theta)] - v_{dy} \cos \theta \sin \phi \\ a &= \dot{w} \\ \dot{\theta} &= 0 \\ \dot{w} &= 0 \\ \dot{w}_d &= 0 \end{aligned} \tag{8}$$

When the target has an initial angular velocity $w = 0$, the expressions change to [8]

$$\bar{x}_{i+1} = x_i + v \cos\theta \cdot \Delta T - v_{dx} \cos\theta \cos\phi \cdot \Delta T$$
$$\bar{y}_{i+1} = y_i + v \sin\theta \cdot \Delta T - v_{dx} \cos\theta \sin\phi \cdot \Delta T$$
$$v = 0$$
$$\theta = 0 \tag{9}$$
$$w = 0$$
$$w_d = 0$$

The EKF performs the filtering in a bi-dimensional plane formed by the intersection of the range of the LiDAR and Radar sensor to achieve a three-dimensional reconstruction of the sensed target and a rotation is accomplished on the *x*-axis of the sensors, using the cylindrical coordinates as orientation. Figure 4 shows the dynamics between the UAV for generating a three-dimensional reconstruction from bi-dimensional data gathered by the sensors and the target in the *XYZ* plane.

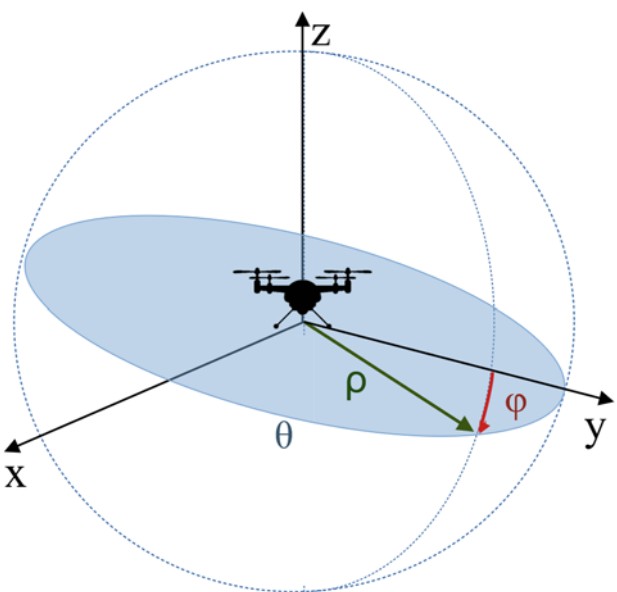

**Figure 4.** Three-dimensional reconstruction with UAV.

For the data fusion design, the RPLIDAR Slam S1 LiDAR sensor and the Positio2go BGT24MTR12 Radar were used as references. The LiDAR sensor operates in 2D with rotation capability, delivering data for a 360° scan, and the Radar achieves a range of 10 m. The range of the sensors according to the implementation of data fusion in the UAV is shown in Figure 5.

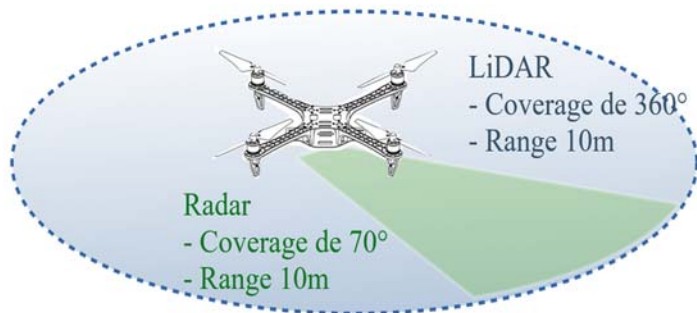

**Figure 5.** LiDAR and Radar sensor range.

Now, to improve the estimation of the measured variable from the noisy sensors, the Kalman filter is implemented through sequential steps: (i) the estimation or prediction of the system behavior from the nonlinear equations; (ii) the calculation of the Kalman

gain to reduce the error of the prediction of the current state versus the previous state, and (iii) the update of the measurement matrix, as well as the covariance associated with the uncertainty of the system. The representation from the selected state variables corresponds to the following equation:

$$\dot{x} = f(x) + w \tag{10}$$

where $\dot{x}$ is the vector of the system states and $f(x)$ is a nonlinear function of the states. This state–space model of the system allows us to determine the future states and the output is obtained by filtering the input signal. The Kalman filter performs estimations and corrections iteratively, where the possible errors of the system will be reflected in the covariance values present between the measured values and the values estimated by the filter. The forward projection of the covariance error has the following representation:

$$M_k = \Phi_k P_k \Phi_k^T + Q_k \tag{11}$$

The system update is implemented according to the following equations:

$$K_k = M_k H^T (H M_k H^T + R_k)^{-1} \tag{12}$$

$$P_k = (I - K_k H) M_k \tag{13}$$

$$\bar{z}_k = H\bar{x} + V_k \tag{14}$$

$$\bar{x}_k = \Phi_k \bar{x}_{k-1} + K_k (z_k - H\Phi_k \bar{x}_{k-1}) \tag{15}$$

Based on these state variables, the fundamental matrix for the extended Kalman filter is calculated and must satisfy the condition

$$F = \left. \frac{\delta f(x)}{\delta x} \right|_{x=\hat{x}} \tag{16}$$

$$\Phi_k = I + F \cdot \Delta T \tag{17}$$

Making $\alpha = \Delta T + \theta$, $\beta = -sin\theta + sin\alpha$, and $\chi = -cos\theta + cos\alpha$ in Equation (18), the fundamental matrix is

$$\Phi_k = \begin{bmatrix} 1 & 0 & \frac{\beta}{w} & \frac{v}{w}\chi & \frac{\Delta Tv}{w}\left[cos\,\alpha - \frac{v}{w^2}\beta\right] & -v_d sin\theta cos\Phi \\ 0 & 1 & \frac{-\chi}{w} & \frac{v}{w}\beta & \frac{\Delta Tv}{w}\left[sin\,\alpha + \frac{v}{w^2}\chi\right] & v_d cos\theta sin\Phi \\ 0 & 0 & 1 & \Delta T & 0 & 0 \\ 0 & 0 & 0 & 1 & 0 & 0 \\ 0 & 0 & 0 & 0 & 1 & 0 \\ 0 & 0 & 0 & 0 & 0 & 1 \end{bmatrix} \tag{18}$$

When the angular velocity of the target is zero, the fundamental matrix reduces to the following matrix:

$$\Phi_k = \begin{bmatrix} 1 & 0 & cos\,\theta \cdot \Delta T & -v \cdot sin\theta \cdot \Delta T & 0 & -v_d sin\theta cos\phi \\ 0 & 1 & sin\theta \cdot \Delta T & v \cdot cos\,\theta \cdot \Delta T & 0 & v_d cos\theta sin\phi \\ 0 & 0 & 1 & 0 & 0 & 0 \\ 0 & 0 & 0 & 1 & 0 & 0 \\ 0 & 0 & 0 & 0 & 1 & 0 \\ 0 & 0 & 0 & 0 & 0 & 1 \end{bmatrix} \tag{19}$$

The matrix associated with the system noise $Q_k$ was calculated from the discrete output matrix $G_k$. The matrix must consider the output variables on which the Kalman filter can act. For the proposed model, the angular acceleration of the target has been taken into account, as well as the angular velocity and acceleration of the UAV. The output matrix $G_k$ for the EKF is presented below.

$$G_k \mu = \int_0^{Ts} \Phi_k(\tau) \cdot G \cdot d(\tau) \tag{20}$$

$$G_k \mu = \begin{bmatrix} \frac{\Delta T^2}{2} \cos\theta & 0 & -\cos\theta\cos\phi \cdot \Delta T \\ \frac{\Delta T^2}{2} sin\theta & 0 & -\cos\theta\sin\phi \cdot \Delta T \\ \Delta T & 0 & 0 \\ 0 & 0 & 0 \\ 0 & \Delta T & 0 \\ 0 & 0 & \Delta T \end{bmatrix} \cdot \begin{bmatrix} \mu_a \\ \mu_w \\ \mu_{wd} \end{bmatrix} \tag{21}$$

The noise matrix from the output matrix is calculated through the following expression:

$$Q_k = G_k \cdot E[\mu \cdot \mu^T] \cdot G_k^T \tag{22}$$

where

$$E\left[\mu \cdot \mu^T\right] = \begin{bmatrix} \sigma_a^2 & 0 & 0 \\ 0 & \sigma_w^2 & 0 \\ 0 & 0 & \sigma_{wd}^2 \end{bmatrix} \tag{23}$$

With $\gamma = cos\theta\ cos\varphi$, $\nu = cos\varphi\ sin\varphi$, $\eta = sin\theta\ sin\varphi$, $\kappa = sin\theta\ cos\varphi$, $\sigma = cos\theta\ sin\varphi$, $\lambda = cos\varphi\ sin\varphi$, the noise matrix is defined as

$$Q_k = \begin{bmatrix} \left(\frac{\Delta T^2}{2}\sigma_a \cos\theta\right)^2 + (\Delta T \cdot \sigma_{wd}\gamma)^2 & \frac{\Delta T^4}{4}\sigma_a{}^2\kappa + (\Delta T \cdot \sigma_{wd}\cos\theta)^2\lambda & \frac{\Delta T^3}{4}\sigma_a{}^2\kappa & 0 & 0 & -(\Delta T \cdot \sigma_{wd})^2\gamma \\ \left(\frac{\Delta T^2}{2}\sigma_a sin\theta\right)^2 + (\Delta T \cdot \sigma_{wd}\cos\theta)^2\nu & \frac{\Delta T^4}{4}\sigma_a{}^2\kappa + (\Delta T \cdot \sigma_{wd}\sigma)^2 & \frac{\Delta T^3}{2}\sigma_a{}^2 sin\theta & 0 & 0 & -(\Delta T \cdot \sigma_{wd})^2\eta \\ \frac{\Delta T^3}{2}\sigma_a{}^2 \cos\theta & \frac{\Delta T^3}{2}\sigma_a{}^2 sin\theta & \Delta T^2\sigma_a^2 & 0 & 0 & 0 \\ 0 & 0 & 0 & 0 & 0 & 0 \\ 0 & 0 & 0 & 0 & \frac{\Delta T^2}{2}\sigma_{\dot{w}} & 0 \\ -(\Delta T \cdot \sigma_{wd})^2\gamma & -(\Delta T \cdot \sigma_{wd})^2 \cos\theta\sin\phi & 0 & 0 & 0 & -(\Delta T \cdot \sigma_{wd})^2 \end{bmatrix} \tag{24}$$

Regarding the variables obtained from the sensors, it should be taken into account that the LiDAR and Radar sensors provide the measurements in different formats. For the LiDAR case, position data are retrieved in rectangular coordinates for $x$ and $y$ that correspond to the first two variables of the state vector. Since $x_k$ has six state variables, the measurement matrix for LiDAR data processing should operate only on the $x$ and $y$ variables, making the product between the state vector $x_k$ and $H$, conformable, i.e.,

$$H_L = \begin{bmatrix} 1 & 0 & 0 & 0 & 0 & 0 \\ 0 & 1 & 0 & 0 & 0 & 0 \end{bmatrix} \tag{25}$$

For Radar, the measurement matrix changes to

$$H_R = \begin{bmatrix} 1 & 0 & 0 & 0 & 0 & 0 \\ 0 & 1 & 0 & 0 & 0 & 0 \\ 0 & 0 & 1 & 0 & 0 & 0 \\ 0 & 0 & 0 & 1 & 0 & 0 \\ 0 & 0 & 0 & 0 & 1 & 0 \end{bmatrix} \tag{26}$$

The measurement error covariance matrix of the LiDAR sensor, obtained from the statistical analysis of the dataset obtained from this sensor, is as follows:

$$R_{k-L} = \begin{bmatrix} 0.0222 & 0 \\ 0 & 0.0222 \end{bmatrix} \tag{27}$$

$R_{k-L}$ is obtained from the variance in the LiDAR dataset. Likewise, the measurement error covariance matrix of the Radar sensor obtained is

$$R_{k-R} = \begin{bmatrix} 0.088 & 0 & 0 & 0 & 0 \\ 0 & 0.00088 & 0 & 0 & 0 \\ 0 & 0 & 0.088 & 0 & 0 \\ 0 & 0 & 0 & 0.0088 & 0 \\ 0 & 0 & 0 & 0 & 0.08 \end{bmatrix} \tag{28}$$

These covariance values are directly related to the resolution and reliability of both the LiDAR and the Radar sensors. In the LiDAR case, the uncertainty is present in its measurement of the target distance, measured and represented as $x$ and $y$ coordinates, while for the Radar, this uncertainty is found in this same measurement, but is represented as a vector distance of magnitude $\rho$ and angle $\theta$. Likewise, the covariance matrix for the Radar includes the estimated velocity at the target.

## 3. Results and Discussion

The EKF filter was implemented under numerical evaluation using Matlab®. To determine its performance, a dataset combining position measurements from a LiDAR and Radar sensor for a pedestrian and real position measurements for the pedestrian were used, and with these results, an estimation of the performance was obtained using the RMSE [12]. To evaluate the robustness of the model, the dataset was contaminated with different levels of additive white Gaussian noise (AWGN).

The system was initialized by predefining values for the state matrix as shown in Figure 6, as well as the fundamental matrix, the system covariance matrix, and the noise matrix. Each new LiDAR or Radar sensor input triggers the filtering, starting by determining the time-lapse DT concerning the previous measurement. Next, the state matrix is estimated using the set of Equations (8) or (9) when $w = 0$, the fundamental matrix according to Equations (18) or (19) if $w = 0$, the noise matrix given by (24), and the system covariance matrix as given by Equation (11). Next, the configuration of the measurement and uncertainty matrices, (25) and (27) for the LiDAR case and (26) and (28) for the Radar case, is performed. The Kalman gain given by Equation (13) is determined, and, finally, an updating of the measurement matrix given by (15), as well as the system and state covariance matrix, is achieved.

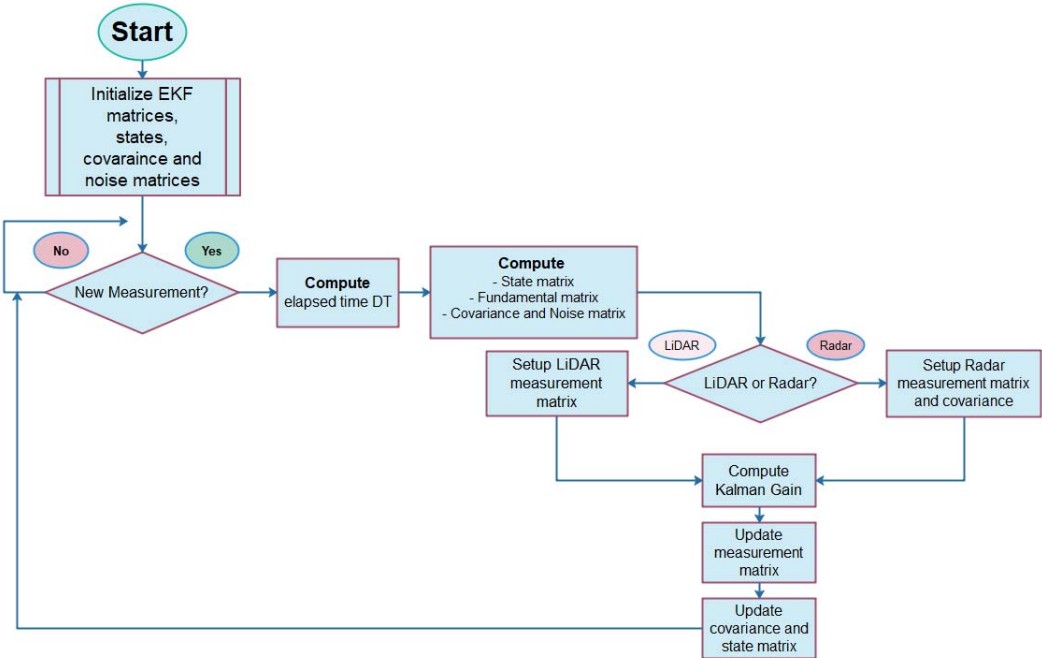

**Figure 6.** Schematic for EKF implementation.

The CTRV model proposed by [8] in the context of autonomous driving was designed, taking as reference highways and locations commonly used by automotive vehicles. In these scenarios, the tangential velocity changes to the sensors are presented to a lesser extent concerning the same scenario, but with measurements taken from a UAV. This behavior is accentuated when it is necessary to perform a three-dimensional reconstruction of the moving target. The CTRV model developed in this work includes the angular velocity of the drone, modifying the fundamental matrix of the system, as well as the noise matrix associated with the system, and a favorable response of the filter to the newly established changes was observed.

To evaluate the response of the *x* and *y* position variables to measurements contaminated with noise, the equations of the CTRV model were implemented in Matlab®, and a sweep of the position variables contaminated with AWGN was performed. The response of the *x* and *y* state variables of the EKF to these contaminated measurements is shown in Figure 7.

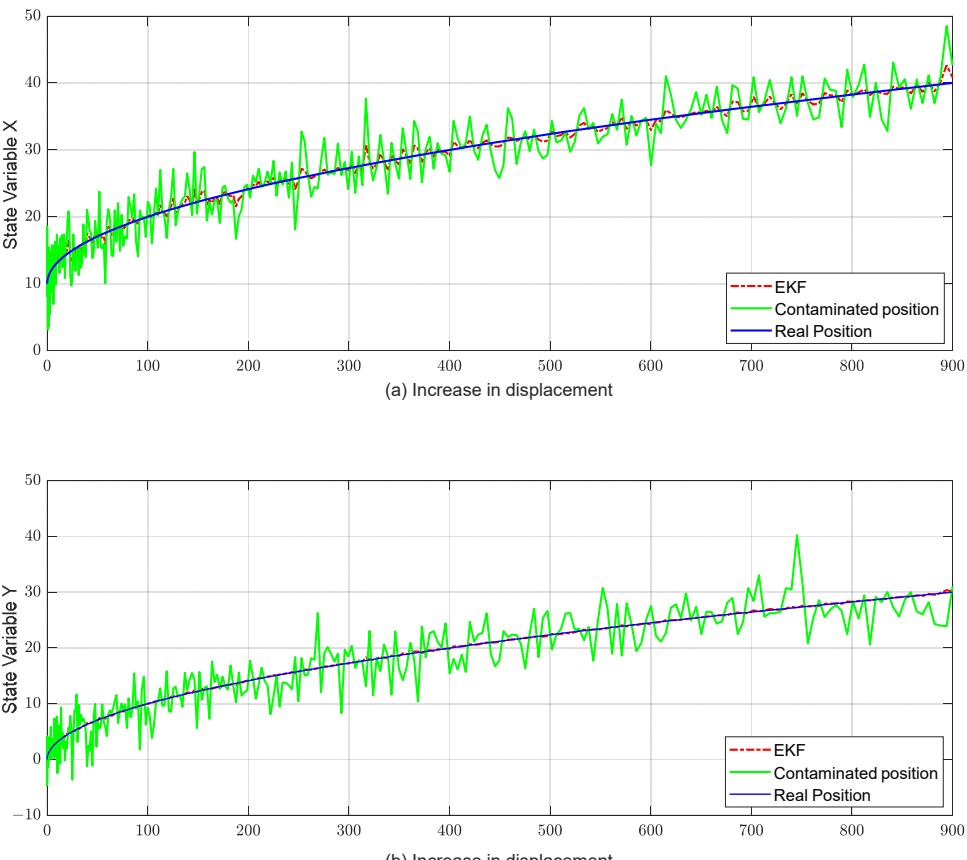

**Figure 7.** Response of the EKF in the state variables (**a**) *x* and (**b**) *y* to measurements contaminated with AWGN. Source: authors.

The response of the EKF to significant changes in the angular velocity of the target, represented as a change in the direction of the trajectory on the *x*-axis, is presented below. The EKF succeeds in predicting the target (green band), even at the point of greatest deflection. The zoom of the filter's response to the change in trajectory is shown in Figure 8. The EKF was tested with the help of a dataset that provides 1225 positions and angles from simultaneous Radar and LiDAR measurements, where the Radar sensor provides the distance along with the angle of displacement, concerning the horizontal of the Radar, and also the angular velocity detected by the Radar; the LiDAR sensor gives the position in x and y coordinates. This dataset was contaminated with AWGN by increasing the noise power progressively and testing the EKF.

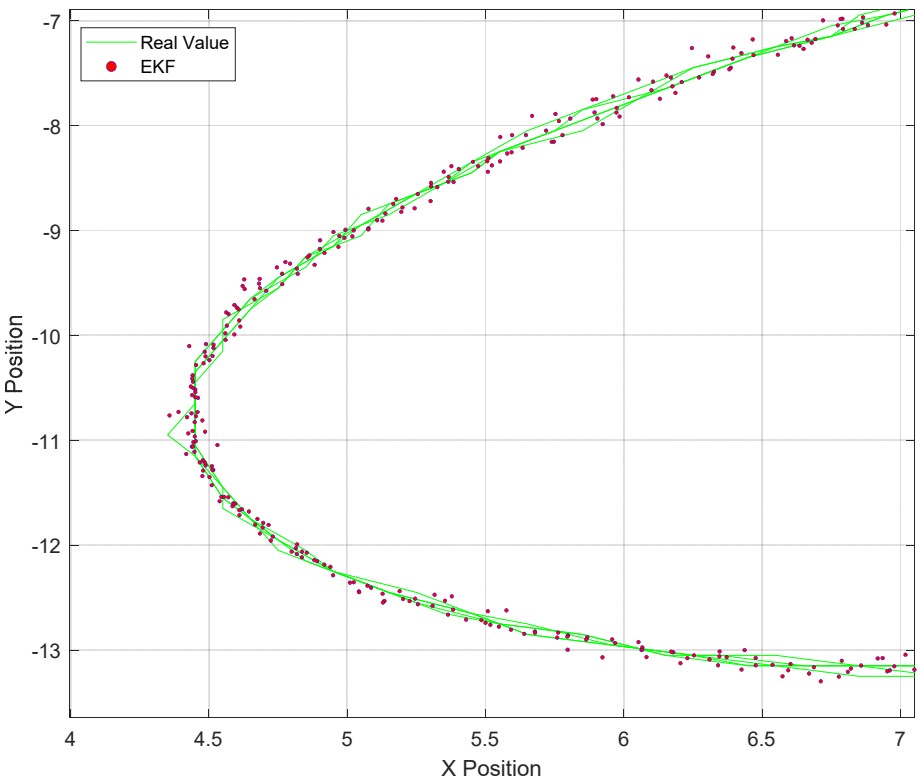

**Figure 8.** EKF response to trajectory changes in the moving target.

The representation of the real data versus the measured data from the Radar and LiDAR sensors is visualized in Figure 9.

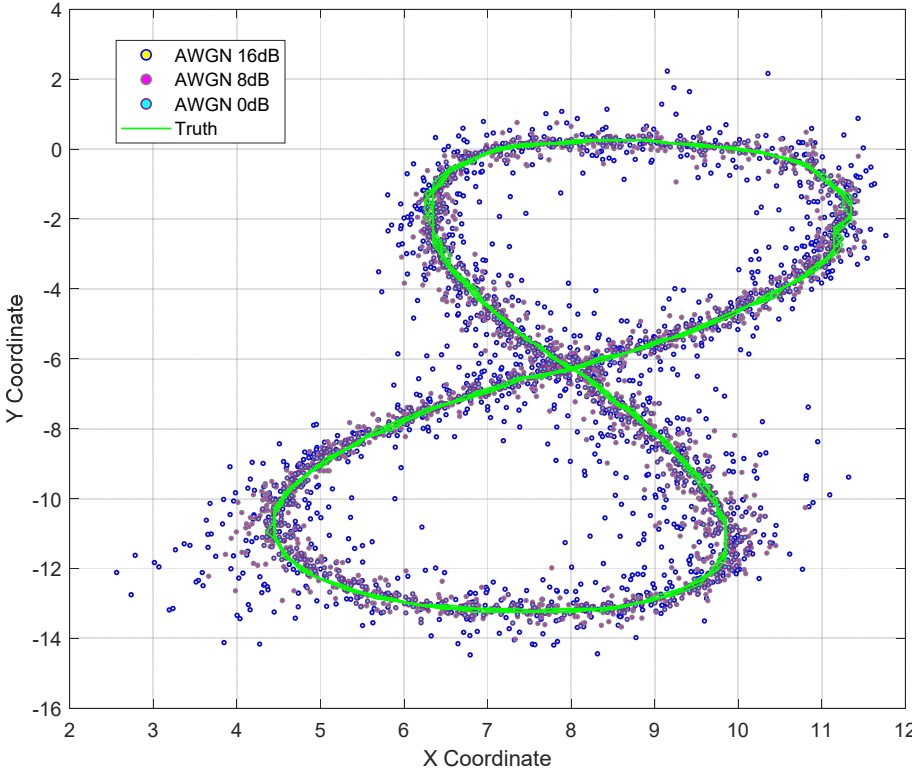

**Figure 9.** The plot of x and y position values for a 16 dB AWGN-contaminated LiDAR and Radar fusion.

To draw a comparison with other fusion models with Radar data, the root mean square error (RMSE) was computed, and it was possible to validate the effectiveness of the proposed CTRV model against fusions of previously used sensor data. The root mean square error is reduced from 0.21 to 0.163 in terms of the linear model; however, in contrast to the state-of-the-art [13] the unscented Kalman filter (UKF) maintains a better response compared to EKF based on the CTRV model. The response of the EKF to AWGN variations in the input data is presented below.

Figure 10 shows that the EKF acts by reducing the difference between the real values (red signal) and the values contaminated with Gaussian noise (blue signal). In [13], the authors state that the RMSE response can be improved with the unscented Kalman filter; however, it should be noted that this filter implies a higher computational complexity concerning the EKF. On the other hand, the response of the KF with a linear model and increasing AWGN is shown in Figure 11.

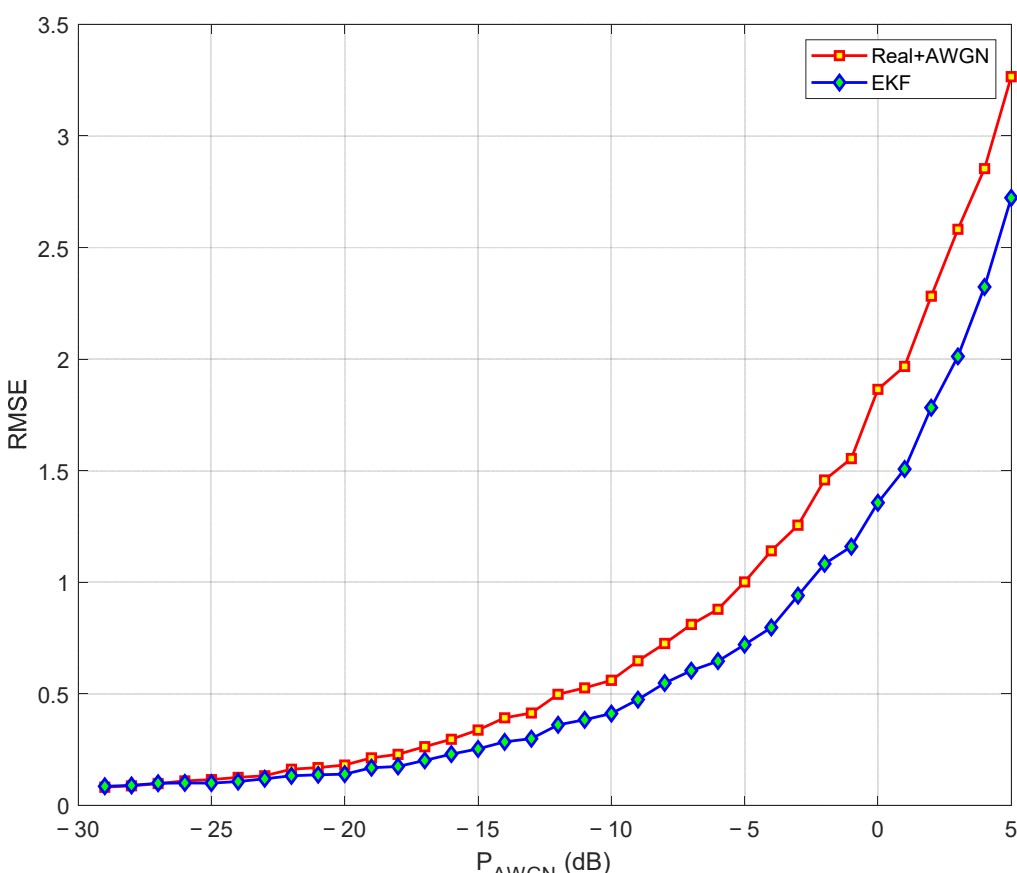

**Figure 10.** RMSE vs. AWGN in CTRV model.

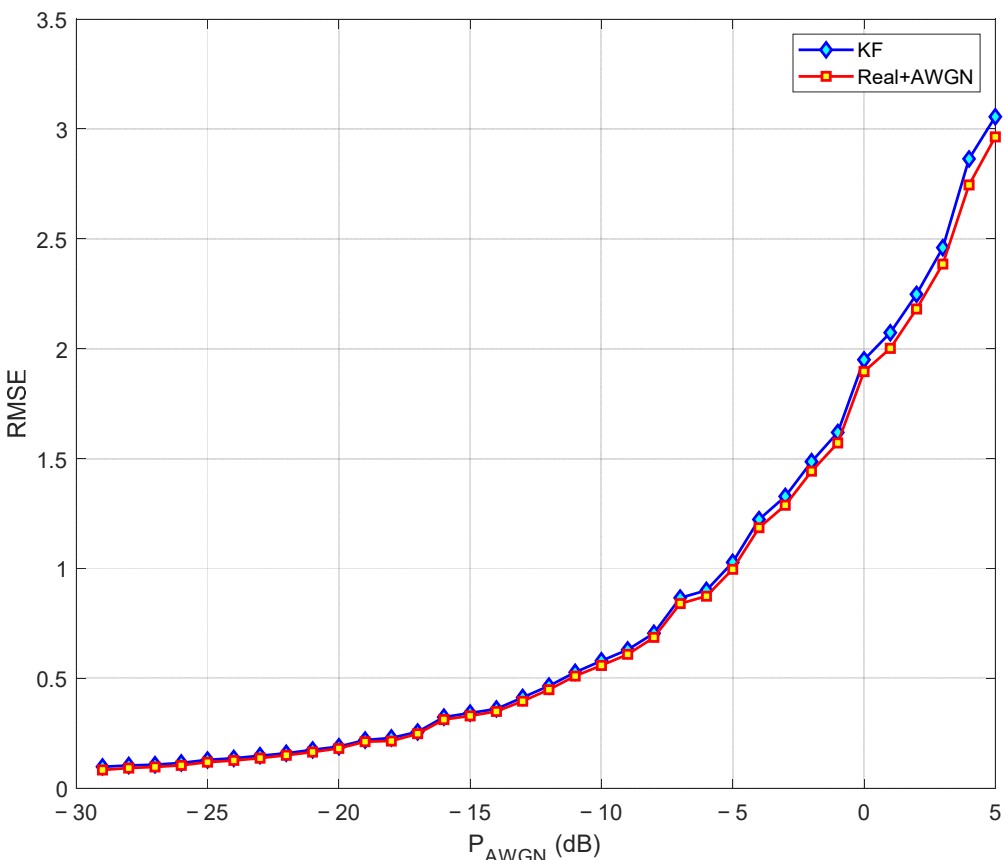

**Figure 11.** RMSE vs. AWGN in the linear model.

## 4. Conclusions

An architecture for LiDAR and Radar sensor data fusion through the extended Kalman filter model was implemented based on the CTRV model and the angular velocity projection of a UAV (a parameter not identified in related previous research). The robustness against trajectory changes for a moving target was demonstrated and determined by the angular velocity and angle of the target concerning the UAV provided by the LiDAR and Radar sensor. The evaluation of this model from the dataset allowed an accurate tracking of the target in the face of position changes. In CTRV modeling, the angle of radar and angular velocity of drone, when working together ensure a better response of the EKF. In the review of the state of the art, no references have been found that include the angular velocity of the drone. The projection of the UAV angular velocity on an *xz*-plane allows a bi-dimensional analysis to be performed, as well as the modeling of the drone–moving target system, without negatively affecting the EKF response.

The CTRV model proposed in this article for the drone–moving target system was validated by numerical analysis using real data captured from LiDAR and Radar sensors. In future work, when the implemented system in a UAV including the kinematic proposed model requires a performance validation of the data sensor fusion using the EKF, the implementation of the EKF algorithm must be evaluated in a Field-Programmable Gate Array (FPGA) or System-on-Chip module due to their parallel processing capacity.

**Author Contributions:** Conceptualization, O.J.M., E.A.F. and M.J.S.; methodology, O.J.M.; software, O.J.M. and E.A.F.; validation, O.J.M., E.A.F. and M.J.S.; formal analysis, O.J.M.; investigation, O.J.M. and E.A.F.; resources O.J.M.; writing—original draft preparation, O.J.M., E.A.F. and M.J.S.; writing—review and editing, O.J.M. and E.A.F.; visualization, E.A.F. and M.J.S.; supervision, E.A.F.; project administration, E.A.F. All authors have read and agreed to the published version of the manuscript.

**Funding:** This study was supported by the UPTC SGI 3139 Clarifier Research Project, as a partner of an International Research Cooperation agreement under the NATO Science for Peace Program—SPS G5888.

**Data Availability Statement:** The data presented in this study are available on request from the corresponding author.

**Acknowledgments:** All authors would like to sincerely thank the reviewers and editors for their suggestions and opinions for improving this article.

**Conflicts of Interest:** The authors declare no conflict of interest.

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
