# Peer review of "Application of Data Sensor Fusion Using Extended Kalman Filter Algorithm for Identification and Tracking of Moving Targets from LiDAR–Radar Data"

_remotesensing, doi:10.3390/rs15133396_

Round 1

Reviewer 1 Report

please see attached PDF document for comments and recommendations

english language is good, no major issues

Reviewer 2 Report

My recommendation for this paper is “Major revision”. The following comments should be addressed carefully before my further decision.
(1) The reviewer considered that the contribution of this work is limited. Compared with Ref[12], the author only took the
angle θ formed between the radar and the target for consideration.

(2) The author mentioned that “The UAV dynamics was obtained from the 2D CTRV model [8] for vehicle detection on highways. ” in section 2, while in section 3, the author mentioned that “a dataset combining position measurements from a Lidar and Radar sensor for a pedestrian and real position measurements for the pedestrian were used”. The two statements are inconsistent.
(3) From Fig 7(EKF response), one can find that the prediction of variable Y is closer to the actual position than X. The reviewer curious about this. Could the author please explain the reason or principle.

(4) One thing the reviewer want to conform is whether the model and experiments in this paper are for the single target case, and if so, how about the performance if it is applied to the multi-object field. I personally think this is worth exploring, after all sensor fusion, KF is widely used in multi-object detection/tracking.

(5) What is meaning of senθ in EQ9 and 18? Also, please double check the symbols in the equations. The authors should be rigorous and precise

(6) The resolution of the figure is very poor.

(7) What is the meaning of “Source: Authors.” in the caption of the figures. 

(1) Please revise the figures as they are unsharpness.

(2) Please double-check the language and the formulas.

Reviewer 3 Report

Summary:

The paper presents a sensor data fusion algorithm using Extended Kalman Filter (EKF) to track moving targets. It employs Lidar and Radar data sensors to measure position of moving targets and use constant turn and rate velocity (CRTV) kinematic model to identify and track moving targets. The performance of the model is evaluated by measuring root mean square error.

Major Comments:

  • Model description and data collection are confusing.

  • The authors need to simplify model descriptions. e.g. the paper uses CRTV Kinetic model as a reference and hence needs to explain basics of the model. 

  • It is not clear, what is the data collected from moving targets and used for identification and tracking? Is it position, velocity, time, angular velocity or anything else? Data collection should be explained clearly.

  • The authors should present a comparison of the performance of the presented algorithm against other existing data fusion models.

 Minor Comments:

  • Few sentences in the article do not make any sense and need to be rewritten. e.g. “measurement from their type of domain (electrical or optical, for example), with higher reliability from jointly information from multiple sensors, eventually heterogeneous, that make inferences that cannot be possible from a single sensor”, in line 45 of page 2 is  not understandable. 

  • There are many typo error e.g. “meanly” instead of “mainly” is written in line 58 of page 2.

Minor Comments:

  • Few sentences in the article do not make any sense and need to be rewritten. e.g. “measurement from their type of domain (electrical or optical, for example), with higher reliability from jointly information from multiple sensors, eventually heterogeneous, that make inferences that cannot be possible from a single sensor”, in line 45 of page 2 is  not understandable. 

  • There are many typo error e.g. “meanly” instead of “mainly” is written in line 58 of page 2.

Author Response

Please see the attachment "Response to reviewer 3".

Round 2

Reviewer 2 Report

The reviewer considers that the authors did not address my concern well. 

Moderate editing of English language required